HCV core antigen is an alternative marker to HCV RNA for evaluating active HCV infection: implications for improved diagnostic option in an era of affordable DAAs

Wasitthankasem Rujipat 1
Vichaiwattana Preeyaporn 1
Auphimai Chompoonut 1
Siripon Nipaporn 1
Klinfueng Sirapa 1
Tangkijvanich Pisit 2
Vongpunsawad Sompong 1
Poovorawan Yong Yong.P@chula.ac.th 1
1 Center of Excellence in Clinical Virology, Faculty of Medicine, Chulalongkorn University , Bangkok , Thailand
2 Research Unit of Hepatitis and Liver Cancer, Department of Biochemistry, Faculty of Medicine, Chulalongkorn University , Thailand
Flores-Valdez Mario Alberto
Electronic publication date: 2017 Nov 6
Publication date: 2017
Volume: 5
Electronic Location ID: e4008
Received 2017 Aug 16; Accepted 2017 Oct 18
Copyright: ©2017 Wasitthankasem et al.
Copyright year: 2017
Copyright holder: Wasitthankasem et al.
License: This is an open access article distributed under the terms of the Creative Commons Attribution License, which permits unrestricted use, distribution, reproduction and adaptation in any medium and for any purpose provided that it is properly attributed. For attribution, the original author(s), title, publication source (PeerJ) and either DOI or URL of the article must be cited.
License URL: https://creativecommons.org/licenses/by/4.0/

Keywords: Hepatitis C virus, HCV Ag, HCV RNA, Active HCV infection

Funding: National Science and Technology Development Agency P-15-5004 Chulalongkorn University Centenary Academic Development Project CU56-HR01 Thailand Research Fund RTA5980008 Ratchadaphiseksomphot Endowment Fund GCE58- 014-30-004 King Chulalongkorn Memorial Hospital This work was supported by the Research Chair Grant from the National Science and Technology Development Agency (P-15-5004), Chulalongkorn University Centenary Academic Development Project (CU56-HR01); the Senior Research Scholar, Thailand Research Fund (RTA5980008); Ratchadaphiseksomphot Endowment Fund for Postdoctoral Fellowship to Rujipat Wasitthankasem; the Center of Excellence in Clinical Virology, Chulalongkorn University (GCE58- 014-30-004); and King Chulalongkorn Memorial Hospital. The funders had no role in study design, data collection and analysis, decision to publish, or preparation of the manuscript.

==============================
The core antigen of the hepatitis C virus (HCV Ag) presents an alternative marker to HCV RNA when screening patients for HCV viremia. This study sought to evaluate the utility of HCV Ag as a marker to assess active HCV infection in individuals residing in an HCV-endemic area. From 298 HCV-seropositive individuals evaluated for the presence of anti-HCV antibody, HCV Ag and HCV RNA, anti-HCV antibody was detected in 252 individuals (signal-to-cutoff ratios ≥5), HCV RNA was detected in 222 individuals (88%), and HCV Ag was reactive (≥3 fmol/L) in 220 individuals (87%). HCV genotype 1, 3, and 6 were identified. HCV Ag significantly correlated with HCV RNA irrespective of HCV genotype and/or HBV co-infection (log HCV RNA = 2.67 + 0.95 [log HCV Ag], R2 = 0.890, p < 0.001). To predict HCV viremia (HCV Ag ≥ 3 fmol/L), the accuracy, sensitivity, specificity, positive predictive value, and negative predictive value were 99%, 99%, 100%, 100% and 97%, respectively. We concluded that HCV Ag was a good surrogate marker for HCV RNA and could be used to diagnose active HCV infection in a resource-limited setting. As a result, a cost-effective strategy for screening and identifying active HCV carriers using HCV Ag detection would enable more patients access to efficacious and increasingly affordable direct-acting antivirals (DAAs) for the treatment of HCV infection.

Introduction

Infection with hepatitis C virus (HCV) can lead to acute or chronic hepatitis, liver fibrosis, cirrhosis, end-stage liver disease, and hepatocellular carcinoma (Perz et al., 2006). Disease progression after HCV infection depends on factors including gender, coinfection with HIV, alcohol consumption, and duration of chronic infection (Hajarizadeh, Grebely & Dore, 2013; El-Serag, 2012). The global population seroprevalence of anti-HCV antibodies is estimated at around 1.6%, or roughly 115 million people (Gower et al., 2014). The presence of HCV antibodies can be found in spontaneous clearance, resolved infection post-treatment or persistently active disease. Early treatment for HCV infection, especially in the acute phase, can facilitate viral clearance and prevent chronic infection, thereby limiting HCV-induced liver damage and mortality (Jaeckel et al., 2001; Gerlach et al., 2003; Singal et al., 2010).

Screening for anti-HCV antibody (anti-HCV Ab) status often facilitates HCV surveillance in the community (Meffre et al., 2010; Garcia Comas et al., 2015; Morisco et al., 2016). Although simple, such an assay cannot differentiate between past and present infection and requires supplemental HCV RNA testing to confirm active HCV infection and monitor antiviral treatment. Despite its high sensitivity and reliability, an HCV RNA assay involving nucleic acid testing (NAT) and quantitative real-time RT-PCR requires skilled laboratory personnel, sophisticated equipment, and expensive reagents (Roth et al., 2012; Halfon et al., 2006). Therefore, routine screening using these tests is limited particularly for many resource-constrained setting. In contrast, testing for HCV core antigen (HCV Ag) presents a more attractive alternative owing to the lower cost and short turnaround time. HCV Ag has been shown to be an indirect marker for HCV replication comparable to the detection of HCV RNA (Schuttler et al., 2004; Bouvier-Alias et al., 2002; Ottiger, Gygli & Huber, 2013; Florea et al., 2014). In addition to serving as a reliable marker to diagnose active HCV infection, HCV Ag can also be used to evaluate the treatment response to antiviral therapy (Mederacke et al., 2009; Rockstroh et al., 2017; Alonso et al., 2017). Another advantage of the HCV Ag assay is that it can often be performed on the same instrument and simultaneously with the anti-HCV assay, an added value when determining the HCV prevalence in the community (Kuo et al., 2012; Mixson-Hayden et al., 2015).

New and effective therapeutic direct-acting antivirals (DAAs) taken orally have become widely available in recent years. DAAs have advanced HCV therapy with higher rates of sustained-virological response (SVR) post-treatment than those obtained from traditional interferon-based therapies independent of HCV genotype (Sulkowski et al., 2014; Zeuzem et al., 2014; Gane et al., 2015). Despite the availability of these highly efficient drugs, asymptomatic nature of HCV infection and expensive diagnostic screening process represent major obstacles in identifying and treating HCV-infected individuals (Cox, 2015). Therefore, a simple, cost-effective testing feasible for routine HCV screening would be ideal for low- to middle- income countries (LMIC) such as Thailand. In this study, we aimed to evaluate the diagnostic utility of HCV Ag as an alternative to HCV RNA to identify active HCV infection in a relatively high endemic area. We also assessed the cost feasibility and savings of implementing such program towards public health policy in an era of DAAs.

Materials and Methods

This follow-up study examined the prevalence and genotypes of HCV found in Petchabun and Khon Kaen province and comprised part of a previous HCV surveillance project involving 3,077 samples from high- and average-level HCV-endemic areas (Wasitthankasem et al., 2017). All HCV-seropositive individuals were informed of their status and invited to participate in confirmatory testing for HCV infection using anti-HCV, HCV Ag, and HCV RNA assays combined with liver enzyme levels and liver stiffness tests in March 2016. The study protocol was approved by the institutional review board of the Faculty of Medicine, Chulalongkorn University (IRB No. 258/58). Objectives of the study were explained to the patients and written informed consents were obtained.

Study subjects

Of the 310 eligible individuals with positive anti-HCV Ab results, 270 subjects participated in the follow-up study (Fig. 1). Another 28 subjects with anti-HCV positive status established after blood donor screening were also enrolled in this study. In all, 298 subjects (aged 34–64 years) provided demographic information and blood samples. Liver function data including aspartate aminotransferase (AST) and alanine transaminase (ALT) levels were obtained, and HCV viral loads were determined from blood plasma. The degree of liver fibrosis was assessed by transient-electrography (FibroScan, Echosens, Paris, France). Hepatitis B virus (HBV) and HIV status were previously determined using HBsAg and HIV Ag/Ab automated screening (ARCHITECT, Abbott Diagnostics, Wiesbaden, Germany).

Figure 1 Evaluation of 298 anti-HCV reactive samples in this study.

Anti-HCV reactive samples were collected from individuals who returned to the HCV follow-up study for the treatment program (270 from the previous HCV study and 28 from self anti-HCV screening) (Wasitthankasem et al., 2017). Anti-HCV nonreactive samples were excluded, and reactive samples were tested for liver function enzymes, liver stiffness, HCV RNA level and HCV Ag level. Viral genotypes were examined in samples with HCV RNA ≥ 12 IU/mL. AST, aspartase aminotransferase; ALT, alanine transaminase.

Anti-HCV serological test

All sera were tested for an anti-HCV Ab using automated chemiluminescent microparticle immunoassays (CMIA) (ARCHITECT anti-HCV assay; Abbott Diagnostics, Wiesbaden, Germany). Signal-to-cutoff ratio (S/CO) indicated the presence of anti-HCV, with S/CO ≥ 1.00 automatically assigned as reactive by the ARCHITECT i1000SR platform. The US Centers for Disease Control and Prevention (CDC) recommended the true predictive result above 95% when the S/CO ratio is ≥5. Therefore, the latter was adopted, and reactive anti-HCV samples were further categorized based on this criterion.

HCV RNA measurement

Plasma HCV RNA level was determined using an automated in vitro reverse-transcription polymerase chain reaction (RT-PCR) assay (Abbott RealTime HCV assay; Abbott Molecular, IL, USA). The lower and upper limits of quantitation of this kit were 12 and 100,000,000 IU/mL, or log 1.08 and log 8.00 IU/mL, with a linearity range between 8.21 log IU/mL to 0.91 log IU/mL (according to the manufacturer’s instructions). Samples over the upper detection limit were examined for viral genotype and were not included in further analyses. Samples with HCV RNA level < 12 IU/mL and ≥12 IU/mL were defined as negative and positive for HCV RNA, respectively.

HCV Ag measurement

HCV core antigen (HCV Ag) quantitation was determined by automated assays using an Architect i2000SR system (ARCHITECT HCV Ag; Abbott Diagnostics, Wiesbaden, Germany). This assay is a CMIA based on the interaction between monoclonal anti-HCV and HCV Ag. A HCV Ag concentration < 3 fmol/L was interpreted as nonreactive, and the other samples with higher titers were interpreted as being reactive for HCV Ag. The detection limit ranged from 0 to 20,000 fmol/L. Therefore, samples with HCV Ag over the limit of detection (20,000 fmol/L) were further diluted (by a factor 1:9), and re-examined by automated dilution protocol in the Architect i2000SR. The machine automatically calculated and reported the final HCV Ag concentration.

HCV genotyping

Genotype was determined based on the nucleotide sequence of the HCV core region. HCV RNA was extracted, and nested RT-PCR of the partial core region was performed on the samples with HCV RNA ≥ 12 IU/mL. Primer pairs of 954F/410 R were used in the first round, and 953F/951R were used in the second round, as previously described (Wasitthankasem et al., 2015). Target PCR amplicon of the core region was purified (ExpinGel SV; GeneAll Biotechnology, Seoul, Korea) and sequenced.

Data and statistical analysis

Continuous variables such as the level of AST and ALT were described in particular groups with <30 U/mL and ≥ 30 U/mL, whereas liver stiffness was categorized according to the Metavir score in which F0-F2 equals liver stiffness <9.5 kPa and F3-F4 equals liver stiffness ≥9.5 kPa (Castera, Forns & Alberti, 2008). The difference between groups and group means was evaluated by Chi-squared test and one-way ANOVA using a Bonferroni model, respectively. Association between the level of HCV RNA and HCV Ag was analyzed in log scale by a linear regression model. Differences of regression data among genotypes were analyzed by general linear model and univariate analysis. The regression model evaluated the odds ratios and 95% confidence intervals of demographic information associated with HCV RNA status. A p-value < 0.05 was considered statistically significant. Sensitivity, specificity, positive predictive value (PPV), negative predictive value (NPV), and accuracy of the diagnostic tests for anti-HCV and HCV Ag markers were calculated using HCV RNA status as a gold standard. All statistical analyses were performed using SPSS version 11.5 for Windows (SPSS, Chicago, IL, USA).

Results

Demographic information of participants

The study design is shown in Fig. 1. From 298 individuals in this study, 290 were reactive to anti-HCV (S/CO ≥ 1.0). There were 242 (83.4%) males and 48 (16.6%) females (mean age 50.4 ± 7.2 and 48.4 ± 8.4 years, respectively). Many of the anti-HCV reactive individuals had elevated levels (≥30 U/mL) of AST (65.5%), ALT (51.7%) and advanced liver fibrosis (47.1%, Metavir score: F3-F4) (Table 1). Among the anti-HCV reactive individuals, 13 of 266 with available HBV status were found reactive to HBsAg, while one was reactive to HIV Ag/Ab.

Table 1 Demographic data of individuals with anti-HCV positive serology in the cohort.

Samples were categorized according to HCV RNA status.

	HCV RNA −ve (N = 68)	HCV RNA +ve (N = 222)	Total (N = 290)	Odds ratio (95% CI)	P-value	
Sex (M:F)	50:18	192:30	242:48	2.3 (1.2, 4.5)	0.013	
Age (Mean ± SD)	50.4 ± 7.7	50.0 ± 7.3	50.1 ± 7.4	0.9 (0.6, 1.3)	0.687	
AST (%)				8.6 (4.7, 16.0)	<0.001	
<30 U/mL	49 (72.1%)	51 (23.0%)	100 (34.5%)			
≥30 U/mL	19 (27.9%)	171 (77.0%)	190 (65.5%)			
ALT				7.7 (3.9, 15.1)	<0.001	
<30 U/mL	56 (82.4%)	84 (37.8%)	140 (48.3%)			
≥30 U/mL	12 (17.6%)	138 (62.2%)	150 (51.7%)			
Liver stiffnessa (%)				3.9 (2.1, 7.1)	<0.001	
F0-F2	52 (76.5%)	101 (45.9%)	153 (52.9%)			
F3-F4	16 (23.5%)	120 (54.3%)a	136 (47.1%)			
Notes.

a Liver stiffness (F0-F2, <9.5 kPa; F3–F4, ≥9.5 kPa) could not be determined in 1 individual.

Additional measurements in HCV antibody-positive samples showed the presence (222/290) and absence (68/290) of HCV RNA (Fig. 1). One sample with HCV RNA exceeding the upper limit of detection (more than log 8 IU/mL) was considered RNA positive but not included in further analyses. All but two of the HCV RNA-positive subjects were reactive to HCV Ag (≥3 fmol/L). A majority of samples with positive viral RNA showed elevated liver enzymes (AST = 77.0% and ALT = 62.2%) and had severe fibrosis (54.3%) (Table 1). There were significant associations between positive HCV RNA and male gender (OR 2.3, 95% CI [1.2–4.5], p = 0.013), elevated liver enzyme levels (AST; OR 8.6, 95% CI [4.7–16.0], p < 0.001 and ALT; OR 7.7, 95% CI [3.9–15.1], p < 0.001) and advance liver stiffness (OR 3.9, 95% CI [2.1–7.1], p < 0.001). Genotyping of HCV revealed 61 samples with genotype 1 (subtype 1a = 44 and 1b = 17), 71 samples with genotype 3 (subtype 3a = 69 and 3b = 2) and 90 samples with genotype 6 (subtype 6c = 1, 6f = 72, 6i = 4 and 6n = 13).

Anti-HCV, HCV Ag and HCV RNA status

A schematic diagram of anti-HCV, HCV Ag, and HCV RNA testing in this study is shown in Fig. 2. Thirty-eight out of 290 individuals with anti-HCV S/CO < 5.0 and HCV Ag < 3.0 fmol/L tested negative for HCV RNA. Of the 252 samples with anti-HCV S/CO ≥ 5, 220 samples with HCV Ag ≥ 3 fmol/L were positive for HCV RNA. In 32 samples with anti-HCV S/CO ≥ 5 and HCV Ag < 3 fmol/L, two samples were positive for HCV RNA.

Figure 2 Schematic diagram of all 290 anti-HCV reactive samples.

Anti-HCV reactive samples were categorized based on S/CO ratio = 5. HCV Ag level was determined in all anti-HCV reactive samples and sample with HCV Ag ≥ 3 fmole/L was designated as reactive. HCV RNA level was determined to confirm HCV active infection with the presence of RNA level ≥ 12 IU/mL.

Among 13 HBV reactive samples, there were six HCV RNA positive samples, five of which were positive for HCV Ag. One sample showed evidence of triple infection with HIV, HBV, and HCV.

Correlation of anti-HCV and HCV Ag with HCV RNA level

Examination of the correlations among anti-HCV, HCV Ag, and HCV RNA levels in log scale showed that anti-HCV significantly correlated with HCV RNA (log HCV RNA = −0.882 + 5.492 [log anti-HCV], R2 = 0.534, p < 0.001) (Fig. 3A). Using an anti-HCV S/CO cut-off at 5, there were 30 false-positive (FP) samples with HCV RNA < 12 IU/mL (Table S1) and no false-negatives (FN). Better correlation was observed between HCV Ag and HCV RNA level (log HCV RNA = 2.67 + 0.95 [log HCV Ag], R2 = 0.890, p < 0.001) (Fig. 3B). Taking an HCV Ag cut-off at 3 fmol/L, only two FN with HCV RNA levels of 501 and 1,445 IU/mL were identified (Fig. 3B and Table S1). Both samples had HCV Ag levels of 0.0 and 2.1 fmol/L and were of genotype 3a (Fig. 4B).

Figure 3 Correlation of anti-HCV and HCV Ag with HCV RNA level.

Anti-HCV and HCV RNA level were analyzed in log scale (A). The lower limit of detection line for HCV RNA was 12 IU/mL. The cutoff line of anti-HCV was 5.0 S/CO. HCV Ag and HCV RNA concentration were analyzed in log scale (B). The lower limit of detection line for HCV RNA was 12 IU/mL. The cutoff line of HCV Ag was 3 fmol/L.

Figure 4 Correlation of HCV Ag with HCV RNA concentration for different genotypes.

Comparison of HCV Ag and HCV RNA level was analyzed in log scale for genotype 1 (A), genotype 3 (B) and genotype 6 (C). The lower limit of detection line for HCV RNA was 12 IU/mL. The cutoff line of HCV Ag was 3 fmol/L.

The mean levels of HCV RNA in the samples with genotype 1 (log 6.0 ± 0.9 IU/mL), genotype 3 (log 5.6 ± 1.1 IU/mL), and genotype 6 (log 6.1 ± 0.8 IU/mL) were determined along with the mean HCV Ag (genotype 1, log 3.5 ± 1.0 fmol/L; genotype 3, 2.9 ± 1.1 fmol/L; genotype 6, 3.8 ± 0.7 fmol/L) (Table 2). Genotypes showed significant differences in the log mean of both HCV RNA and HCV Ag (p = 0.001 and p < 0.001, respectively). Samples with HCV genotype 3 had lower levels of HCV RNA and HCV Ag than genotype 1 (p = 0.014 and p = 0.001, respectively) and 6 (p = 0.001 and p < 0.001, respectively). Based on the regression data, HCV Ag showed greatest correlation with HCV RNA in samples with genotype 1 (log HCV RNA = 2.97 + 0.87 [log HCV Ag], R2 = 0.905, p < 0.001) followed by genotype 6 (log HCV RNA = 2.323 + 1.01 [log HCV Ag], R2 = 0.870, p < 0.001), and genotype 3 (log HCV RNA = 2.75 + 0.97 [log HCV Ag], R2 = 0.848, p < 0.001) (Fig. 4 and Table 2). Finally, the slight difference in regression data among genotypes was not statistically significant (p = 0.102).

There was no significant difference in the log mean of HCV RNA and HCV Ag between HCV mono-infection (N = 215, log 5.9 ± 1.0 IU/mL and log 3.4 ± 1.0 fmol/L) and HCV/HBV co-infection (N = 5, log 5.7 ± 1.7 IU/mL and log 3.0 ± 1.5 fmol/L) (p-value > 0.05). HCV Ag highly correlated with HCV RNA in HCV/HBV co-infection (log HCV RNA = 2.44 + 1.08 [log HCV Ag], R2 = 0.968, p = 0.002), while correlation of these two markers between mono-infection and co-infection was not significantly different (p = 0.119). The one sample with triple infection (HCV/HBV/HIV) had viral load and HCV Ag at log 6.7 IU/mL and log 4.10 fmol/L, respectively.

Evaluation of anti-HCV and HCV Ag assay in predicting HCV infection

To identify HCV viremia, an anti-HCV S/CO cut-off at 5 showed 89.7% accuracy, 100% sensitivity, and 55.9% specificity. Positive predictive value (PPV) and negative predictive value (NPV) were 88.1% and 100%, respectively (Table 3). By applying the HCV Ag assay to predict active HCV infection, a cut-off of 3 fmol/L had the greatest accuracy (99.3%), with 100% specificity, 99% sensitivity, 100% PPV, and 97% NPV. Using both anti-HCV and HCV Ag markers (S/CO ≥ 5 and HCV Ag ≥ 3 fmol/L), diagnostic predictions of HCV viremia were similar to those using HCV Ag marker alone (Table S1).

Table 2 Correlation regression between HCV RNA and HCV Ag among different HCV genotypes (in log scale).

	Genotype 1 (N = 61)	Genotype 3 (N = 69)	Genotype 6 (N = 90)	TOTAL (N = 220)a	P-value	
HCV RNA logb IU/mL (Median)	6.0 ± 0.9 (6.32)	5.6 ± 1.1 (5.89)	6.1 ± 0.8 (6.43)	5.9 ± 1.0 (5.87)	0.001	
HCV Ag logb fmol/L (Median)	3.5 ± 1.0 (3.79)	2.9 ± 1.1 (3.35)	3.8 ± 0.7 (4.04)	3.4 ± 1.0 (3.7)	<0.001	
Regression datac					0.102	
R	0.952	0.921	0.933	0.927		
R-square	0.905	0.848	0.870	0.860		
p-value	<0.001	<0.001	<0.001	<0.001		
Notes.

a Of 222 detectable HCV RNA samples, one sample with over limit of detection and one sample with undetectable HCV Ag were excluded.

b HCV RNA and HCV Ag level presented as mean ± SD.

c Correlation between HCV Ag and HCV RNA in log scale.

Table 3 Diagnostic test evaluation of anti-HCV, HCV Ag and combination of anti-HCV and HCV Ag tests to predict HCV active infection in community screening.

	% Sensitivity (95% CI)	% Specificity (95% CI)	% Positive predictive value (95% CI)	% Negative predictive value (95% CI)	% Accuracy (95% CI)	
Anti-HCV (S/CO = 5.0)	100.0 (98.4–100.0)	55.9 (43.3–67.9)	88.1 (83.4–91.8)	100.0 (90.7–100.0)	89.7 (85.6–92.9)	
HCV Ag (3 fmol/L)	99.1 (96.8–99.9)	100.0 (94.7–100.0)	100.0 (98.3–100.0)	97.1 (90.1–99.7)	99.3 (97.5–99.9)	
Anti-HCV (S/CO = 5) plus HCV Ag (3 fmol/L)a	99.1 (96.8–99.9)	100.0 (94.7–100.0)	100.0 (98.3–100.0)	97.1 (90.1–99.7)	99.3 (97.5–99.9)	
Notes.

a Anti-HCV (S/CO ≥ 5) plus HCV Ag (≥3 fmol/L) and others.

Discussion

Timely HCV treatment, especially during the acute phase of infection, has shown promise in multiple HCV genotypes and results in improved prognosis (Jaeckel et al., 2001). Identifying actively infected individuals is therefore important in reducing HCV transmission and disease burden, especially in areas of endemicity. Although, anti-HCV screening is the primary test for diagnosis of HCV infection, supplemental testing is still needed to discriminate between resolved and viremic infection. The results of this study suggest that anti-HCV Ab tests had a fair correlation with HCV RNA and are appropriate for primary screening due to its high sensitivity. Previously, it was suggested that an appropriate cut-off could predict HCV viremia (Kuo et al., 2012). Thus, this study applied a S/CO cut-off at 5.0 for predicting a true antibody-positive result ≥95% recommended by the US CDC (https://www.cdc.gov/hepatitis/hcv/labtesting.htm). Among samples with anti-HCV S/CO ≥ 5, HCV RNA was positive in 88.1% (222 of 252), while those with S/CO < 5.0 were all negative. The latter may have resulted from a false-positive anti-HCV result or an individual with resolved disease and/or low antibody titer. Using this cut-off ratio to predict HCV infection, high diagnostic sensitivity (0.0% FN and 11.9% FP) was noted. When the cut-off ratio was adjusted to 15.0, better predictive values for HCV infection were obtained with excellent PPV (98.8%) and specificity (98.5%), albeit with poor NPV (32.8%) and sensitivity (38.3%) (Table S2). Thus, anti-HCV testing with S/CO at 5.0 was appropriate for the first-line screening.

Unlike anti-HCV Ab, HCV Ag showed excellent diagnostic validity and correlation with HCV RNA. At a cutoff of 3 fmol/L, HCV Ag was a good predictive marker of HCV viremia with 0 (0.0%) FP and 2 (2.9%) FN cases. The validity of HCV Ag testing was better than the anti-HCV marker that had 99.1% sensitivity, 100% specificity, 100% PPV, 97.1% NPV and 99.3% accuracy. It has been suggested that combining anti-HCV and HCV Ag at the appropriate cutoff point would improve the predictive value for HCV viremia, but the validity was similar to HCV Ag alone when combining those two markers (Kuo et al., 2012). In addition, HCV Ag correlates well with RNA levels and may therefore serve as a predictor for response-guided therapy and for monitoring treatment response (Nguyen et al., 2017; Alonso et al., 2017).

The 2 FN cases from the HCV Ag assay had low viral loads and were of genotype 3. These results may be related to the viral level and detection limit of the respective genotypes. Low levels of HCV RNA tended to increase the coefficient of variation and contributed to FN on HCV Ag assays (Aghemo et al., 2016; Ottiger, Gygli & Huber, 2013). Several studies suggest that the lower limit of HCV Ag detection is equivalent to HCV RNA between 1,000 and 5,000 IU/mL, therefore low RNA level in these two samples (501 IU/mL and 1,445 IU/mL) may have contributed to the FN result (Florea et al., 2014; Freiman et al., 2016; Ottiger, Gygli & Huber, 2013). In addition, adequate analytical sensitivity at 3 fmol/L of HCV Ag of genotype 3 required approximately 1,002 IU/mL of HCV RNA, which was higher than for genotype 1 (896 IU/mL) (Ross et al., 2010; Tillmann, 2014). Similar to this cohort, greater variation in genotype 3 samples with low HCV Ag but high HCV RNA was previously reported (Ottiger, Gygli & Huber, 2013; Mederacke et al., 2009). Discordant result found in this study (0.7%, 2/290) could potentially be resolved with a confirmatory RNA testing (e.g., reactive anti-HCV and non-reactive HCV Ag) (Rockstroh et al., 2017).

Figure 5 Optional screening strategy for treatment of HCV infection.

HCV genotyping is an option depends on IFN- or DAAs-based therapy.

A meta-analysis demonstrated that HCV Ag had excellent correlation with viremia despite limitations on the effect of HCV genotypes and HBV or HIV co-infection (Freiman et al., 2016). A significant difference was observed in the log mean of viral RNA and antigen levels among genotypes, but this was not apparent in the correlation between these two markers. Previous studies also reported that HCV Ag of genotype 1, 2, 3 and 4 had good correlation with viral load (Mederacke et al., 2009; Ottiger, Gygli & Huber, 2013; Ross et al., 2010). Our study also showed good correlation between HCV Ag and RNA in HCV/HBV co-infection samples. However, poor correlation in the HBV co-infection group may sometimes occur especially with genotypes other than genotype 1 (Mederacke et al., 2012). This may be attributed to very low or very high concentration of HCV Ag relative to HCV RNA in some samples. Therefore, HCV Ag retesting should be performed in the samples with very low or very high concentrations. Excellent correlation had been reported in HCV/HIV co-infection (Mederacke et al., 2012; Thong et al., 2015), but that information was insufficient in triple infection samples. We found only one sample with HCV/HBV/HIV triple infection and elevated HCV viremia and HCV Ag. Further studies are required to elucidate the correlations in subjects with dual infections and triple infections.

Previous studies examining HCV Ag in HCV infection were conducted in industrialized countries (Mederacke et al., 2009; Mixson-Hayden et al., 2015; Rockstroh et al., 2017; Freiman et al., 2016), while data from LMIC have been scarced (Khan et al., 2017; Freiman et al., 2016). Our finding provided the evidence of the utility of HCV Ag for HCV screening in LMIC endemic for HCV due to the assay’s high diagnostic validity and lower cost than an RNA assay, which potentially could improve access to treatment and care in LMIC (Wang, Lv & Zhang, 2017; Khan et al., 2017).

Currently, the World Health Organization initiated a goal to eradicate viral hepatitis by increasing the proportion of diagnosed persons by up to 90% by the year 2030 (World Health Organization, 2017). This policy will require a substantial investment. Fortunately, affordable and highly efficacious DAA treatment is increasingly accessible (Nebehay, 2017). To enhance the likelihood of eradicating HCV, results from our study and others suggest an alternative screening and diagnostic strategy to treat HCV infection, especially in the era of DAAs (Fig. 5). When screening initially involves anti-HCV Ab and confirmatory HCV Ag assay, followed by HCV RNA testing on HCV Ag-negative samples, a cost-effective algorithm by us demonstrate a considerable reduction, roughly 48% compared to the standard expenditure (standard algorithm; anti-HCV test followed by RNA assay, File S1) and is consistent with that proposed by others (Jülicher & Galli, 2017). This strategy would effectively provide equal diagnostic performance that supports the possibility of large-scale implementation.

In conclusion, our population-based study showed high validity of HCV Ag as a reliable marker for diagnosis of active HCV infection. This marker showed excellent correlation with the viral RNA irrespective of genotypes and HBV co-infection. In addition, HCV Ag can serve as a supplemental marker after anti-HCV testing to reduce the sample number requiring further confirmatory RNA assays. A proposed cost-effective strategy would reduce the financial burden required for national screening and improve access to HCV treatment and care in the era of affordable DAAs.

Supplemental Information

Table S1 Comparison of the anti-HCV, HCV Ag and HCV RNA status in all samples

Click here for additional data file.

Table S2 Diagnostic validity of anti-HCV at different S/CO cut-off value

Click here for additional data file.

File S1 Budget calculation based on data utilized from the results in this study

Click here for additional data file.

Supplemental Information 4 Raw data of clinical parameters

Click here for additional data file.

We would like to thank Ms Supapith Saiyatha and Mr. Chaiwat Thongmai, Phetchabun Provincial Public Health Office, Ms. Napha Thanetkongtong and Dr. Viboonsak Vuthitanachot, Chumpare Hospital, Chum Phae, Khon Kaen and Ms Saowakon Sochoo, Lomkao Crown Prince Hospital for specimen collection and collaboration.

Additional Information and Declarations

Competing Interests

Author Contributions

Human Ethics

Data Availability

The authors declare there are no competing interests.

Rujipat Wasitthankasem conceived and designed the experiments, performed the experiments, analyzed the data, wrote the paper, prepared figures and/or tables.

Preeyaporn Vichaiwattana, Chompoonut Auphimai and Sirapa Klinfueng performed the experiments.

Nipaporn Siripon contributed reagents/materials/analysis tools.

Pisit Tangkijvanich contributed reagents/materials/analysis tools, reviewed drafts of the paper.

Sompong Vongpunsawad reviewed drafts of the paper.

Yong Poovorawan conceived and designed the experiments, analyzed the data, contributed reagents/materials/analysis tools, wrote the paper, reviewed drafts of the paper.

The following information was supplied relating to ethical approvals (i.e., approving body and any reference numbers):

This study was part of the overall research “Prevalence and genotypes of hepatitis C virus in Petchaboon and Khon Kane province as a model for treatment”. The study protocol was approved by the institutional review board of the Faculty of Medicine, Chulalongkorn University (IRB No. 258/58).

The following information was supplied regarding data availability:

The raw data has been provided as a Supplemental File.

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
