# Peer review of "HCV core antigen is an alternative marker to HCV RNA for evaluating active HCV infection: implications for improved diagnostic option in an era of affordable DAAs"

_PeerJ, doi:10.7717/peerj.4008_

## Round 0.1 · original submission · Minor Revisions

Your work received mostly positive reviews; I think you can improve your work by following all comments and suggestions of the reviewers (see below).

Reviewer 1 ·

Basic reporting

English language usage is quite OK, although a review could enrich the manuscript.

Experimental design

In this manuscript, Wasitthankasem et al propose the use of the core antigen of the hepatitis C virus (HCV Ag) as an alternative marker to HCV RNA when screening patients for HCV viremia.

Considering that: “Development of highly sensitive of HCV Ag quantification assays correlates well with and is comparable to HCV RNA(Bouvier-Alias et al., 2002; Ottiger et al., 2013; Florea et al., 2014). Therefore, HCV Ag can be used as a reliable marker for the diagnosis of active HCV infection and the evaluation of response to antiviral therapy (Mederacke et al., 2009; Rockstroh et al., 2017; Alonso et al., 2017).”, what is the originality, or even the main objective of the present work? I understand that authors are presenting specific data from a given human population, but they should keep clear what exactly is the novelty of their approach.

Validity of the findings

The data is OK and the experimental work seems to be well conducted, but while reading the manuscript I got a question: Since authors say that “The results of this study suggest that anti-HCV Ab tests had a fair correlation with HCV RNA and are appropriate for primary screening…” are they proposing HCV Ag testing ONLY as a “primary screening” that should later be confirmed through HCV RNA testing? In this case, the discussion about the costs/time needed to HCV RNA testing makes no sense… The proportion of patients that needed diagnosis confirmation through HCV RNA testing seemed still significant to me. Could the authors discuss that?

Additional comments

Some points can be addressed by the authors:

There are no reasons to include the following phrase “The remaining samples were HCV RNA-negative.” at the Abstract since authors claim that, amongst patients, 222 tested positive for HCV RNA. Actually, at this point, it is not clear if patients negative for HCV RNA are simultaneously negative to HCV Ag and/or to anti-HCV (which seems not be the case since the presented numbers could not totally overlap).
Authors mention that disease progression after HCV infection depends on different factors, but they do not present any follow up of the patients. It would be interesting to have some follow up data in order to actually evaluate the efficacy of the proposed diagnostics.
Authors claim that “All but two of the HCV RNA-positive subjects were reactive to HCV Ag (≥3 fmol/L).”. What happened to these two patients? Were they subjected to the same therapeutics as the positive patients?
Again considering these two patients, at the Discussion it is mentioned that both were HCV genotype 3, but the next paragraph starts saying that “This study demonstrated that HCV Ag showed excellent correlation with viremia irrespective (sic) of the HCV genotype…” In my opinion, these two assertions are conflicting. Please comment on that.
Are the authors planning to evaluate the results of direct acting agents (DAAs) therapy as a potential outcome in the present cohort (controlling for HCV genotypes, for example)?

Reviewer 2 ·

Basic reporting

Wastthanasem R et al report a study performed in 298 HCV-seropositive individuals to test the usefulness of HCV Ag as a marker to access active HCV infection in an HCV endemic region. Their findings suggest that HCV Ag is a good surrogate marker for HCV RNA.

English language and written is clear.

My only suggestions in regards to this work are that the authors include some newest references to improve the discussion and how their results relate to these publications. For example,
a) Hepatitis C virus core antigen assay: an alternative method for hepatitis C diagnosis.
Wang L, Lv H, Zhang G. Ann Clin Biochem. 2017 Mar;54(2):279-285. doi: 10.1177/0004563216661218.
b) Can Hepatitis C Virus Antigen Testing Replace Ribonucleic Acid Polymerase Chain Reaction Analysis for Detecting Hepatitis C Virus? A Systematic Review.
Khan H, Hill A, Main J, Brown A, Cooke G.Open Forum Infect Dis. 2017 May 26;4(2):ofw252. doi: 10.1093/ofid/ofw252.

Also, what is the author's opinion on how this marker could potentially work in patients receiving antiviral treatment in their population setting? (The Role of Hepatitis C Virus Core Antigen Testing in the Era of Direct Acting Antiviral Therapies: What We Can Learn from the Protease Inhibitors. Linh Thuy Nguyen, Emma Gray, Aisling O'Leary, Michael Carr, Cillian F. De Gascun, and Irish Hepatitis C Outcomes Research Network).

Experimental design

No comment

Validity of the findings

No comment

Additional comments

I find this work useful since it provides more insight on the effectiveness of an alternative method for HCV diagnostics in regions or countries that may not afford full-scale HCV RNA testing.

Reviewer 3 ·

Basic reporting

The drafting of the introduction is redundant (second paragraph , line 48-56) with respect to the use of serology as a standard marker for diagnosis and the need for the use of confirmatory techniques, as well as the disadvantages thereof, which is mentioned again in paragraph 4 (lines 71-76).


Due to a suggestion in the following section, I consider that the literature reference in line 87 and 95 is needless in that context, so I suggest replacing its justification in relation to the origin of the samples, which come from endemic regions of the infection, demonstrated in this literature reference (Wasitthankasem et al. 2017) .

References in line 372 and 441 do not have a DOI number.

Experimental design

In materials and methods are not described the evaluation of the combination of Ac and Ag tests to predict active infection. In table 3 , the results of HCV Ag and AntiHCV plus HCV Ag are exactly the same.

The content of the statement of line 101 is a description of a result, not of the methodology, should appear in results only.

Validity of the findings

The title of manuscript refers the implications for improve treatment option but there are only one paragraph in the manuscript that mention therapeutic agents and does not make specific reference to how it impacts the use of the determination of the antigen in the treatment of the patient. Neither in the objetive, experimental design or results does contemplate the dynamics of the marker during the treatment; in the discussion are not discussed the implications that the use of the antigen could have like marker of monitoring in the patients.

There are others articles with the same results that this manuscript and the authors do not let us known which is the importance of repeat this analysis.

Roberto Alonso et al. HCV Core antigen assay as an alternative to HCV RNA quantification: a correlation study for assessment of HCV viremia. Enferm Infecc Microbiol Clin. 2017

H.A. Soliman et al. Significance of the hepatitis C virus core antigen testing as an alternative marker for hepatitis diagnosis in Egyptian patients. European Review for Medical and Pharmacological Sciences. 2015

Magali Bouvier-Alias et al. Clinical utility of total HCV core antigen quantification : a new indirect marker of HCV Replication. Hepatology. 2002

Van Helden J et al. Hepatitis C diagnostics: clinical evaluation of the HCV core antigen determination. Z Gastroenterol. 2014.

Chevaliez S et al. Clinical utility of hepatitis C virus core antigen quantification in patients with chronic hepatitis C. J Clin Virol. 2014.

Hadziyannis E, et al . Is HCV core antigen a reliable marker of viral load? A n evaluation of HCV core antigen automated immunoassay. Ann Gastroenterol. 2013

And the others cited in the manuscript.

The demographic data presented is not discussed and/or referred to as relevant to the purpose of the article. That is, the degree of fibrosis, age, gender or level of enzymes directly affect the determination of the HCV core antigen or what is the purpose of presenting them or having collected them in relation to the objective of the work? For example, in line 167, the associations between HCV RNA and elevated liver enzymes levels and advance liver stiffness are widely accepted clinical results, relevant for the HCV treatment and clinical management but irrelevant for the diagnosis of active HCV infection.

For the same reason I higly recommend removing Figure 1, because it does not present relevant data that are not addressed in figure 2. The objetive of this work is correlates HCV RNA and HCV Ag, so the study universe is samples but not patients. Hence only is necesary known the total numer of Ac HCV positive samples that was used for the study.


In Discussion (lines 274-277), the justification described for the antigenic test refers to cost; should be more sustained with respect to an approximate cost of each of the tests or a percentage of total savings in the test algorithm used for a patient with active HCV.

Because the "Optional Screening Strategy" is not based entirely on the results obtained in this manuscript, it is suggested to give credit to the authors on which it bases its proposal. I sugest a statment like this "...These results together with previously reported evidence allow us to suggest a strategy...". On the other hand, the strategy does not contemplate the genotyping of the virus.

Additional comments

No comment

---

## Round 0.2 · accepted · Accept

I appreciate your taking into account all comments raised by the 3 reviewers. There is only a last one, regarding the use of "LMIC" term in line 66 (introduction) that I call your attention to. I look forward to receiving new manuscripts from you submitted to PeerJ.

Reviewer 1 ·

Basic reporting

Authors adequately answered the questions raised by this reviewer.

Experimental design

The authors clearly stated the novelty of their approach – meaning, a cost and effectiveness comparison of the different methods of HCV infection detection, in an environment characterized by a high prevalence of HCV infection, in a low-income country.

Validity of the findings

No comment

Additional comments

Authors adequately answered the questions raised by this reviewer.
The authors clearly stated the novelty of their approach – meaning, a cost and effectiveness comparison of the different methods of HCV infection detection, in an environment characterized by a high prevalence of HCV infection, in a low-income country.
Also, the Discussion was restructured and the information specifying which patients would be tested by which detection method was added. Specifically, authors included that individuals who possessed anti-HCV antibody would be screened for HCV Ag. Only HCV Ag-negative samples should then be confirmed with HCV RNA-testing.
Although no follow-up data were presented in order to actually evaluate the efficacy of the proposed diagnostics, I understand that this was beyond the scope of this study. Also considering treatment, but now concerning the two patients that tested negative to the HCV Ag, the authors just explained that they were referred to the Thailand National Health Security Office as otherwise positive based on the RNA results (gold standard).
In summary, and as previously mentioned, I believe that the authors adequately answered the questions raised by this reviewer and that the modifications enriched the manuscript.

Reviewer 2 ·

Basic reporting

The authors have improved the manuscript. The English language is clear, references have been updated, and the structure is adequate.

Experimental design

The authors have clearly explained their methodology and work algorithm, which in turn is part of their proposal to screen for HCV using the viral antigen.

Validity of the findings

These points were covered by the authors and corrected for clarity as suggested by the reviewers.

Additional comments

As mentioned in the first review, this work is important because it offers an alternative diagnostic method for HCV screening, thus securing treatment of those who are unaware of their condition in regions of high endemicity.

Reviewer 3 ·

Basic reporting

No comment

Experimental design

No comment

Validity of the findings

The changes in title, introduction and conclusion are adequate and let known which is the importance of this study.

I only sugest that in order to unify the terms, use the term : "low- to middle-income countries (LMIC)" in line 66 (introduction)